# DAXX Is a Crucial Factor for Proper Development of Mammalian Oocytes and Early Embryos

**DOI:** 10.3390/ijms22031313

**Published:** 2021-01-28

**Authors:** Irina Bogolyubova, Dmitry Bogolyubov

**Affiliations:** Laboratory of Cell Morphology, Institute of Cytology of the Russian Academy of Sciences, 4 Tikhoretsky Ave., 194064 St. Petersburg, Russia; dbogol@mail.ru

**Keywords:** mammalian early development, chromatin remodeling, heterochromatin, DAXX, H3.3 chaperones, ATRX, genome integrity

## Abstract

The Death-domain associated protein 6 (DAXX) is an evolutionarily conserved and ubiquitously expressed multifunctional protein that is implicated in many cellular processes, including transcription, cellular proliferation, cell cycle regulation, Fas-induced apoptosis, and many other events. In the nucleus, DAXX interacts with transcription factors, epigenetic modifiers, and chromatin-remodeling proteins such as the transcription regulator ATRX—the α-thalassemia/mental retardation syndrome X-linked ATP-dependent helicase II. Accordingly, DAXX is considered one of the main players involved in chromatin silencing and one of the most important factors that maintain integrity of the genome. In this brief review, we summarize available data regarding the general and specific functions of DAXX in mammalian early development, with special emphasis on the function of DAXX as a chaperone of the histone variant H3.3. Since H3.3 plays a key role in the developmental processes, especially in the pronounced rearrangements of heterochromatin compartment during oogenesis and embryogenesis, DAXX can be considered as an important factor supporting proper development. Specifically, loss of DAXX affects the recruitment of ATRX, transcription of tandem repeats and telomere functions, which results in a decrease in the viability of early embryos.

## 1. Introduction

DAXX—the Death-domain associated protein 6 is a multifunctional protein involved in many cellular processes, including apoptosis, protein stability, and regulation of transcription [1,2,3]. The DAXX protein is a chaperone of the histone variant H3.3 [4,5]. DAXX-dependent deposition of H3.3 and known relations of the DAXX protein with transcription factors, epigenetic modifiers, and chromatin-remodeling proteins including the α-thalassemia/mental retardation syndrome X-linked protein ATRX [6] make DAXX a main player in chromatin silencing, mainly in the pericentromeric areas [7]. There is growing recent evidence that mutations in the *DAXX* gene may be associated with some cancers [8], suggesting DAXX as one of the most important factors in maintaining the integrity of the genome.

The functions of DAXX in oogenesis and early embryogenesis of mammals have not yet been deciphered in detail, despite of the fact that the processes of oocyte development during meiosis and fertilization, as well as the reprogramming of parental genomes in early embryos, are accompanied by pronounced rearrangements of the heterochromatic compartment [9,10], in which deposition of H3 variants plays an important role [11]. Accordingly, the DAXX protein may serve as an important factor for the success of developmental processes as an H3.3 chaperone, but the data on this issue are relatively scarce and poorly systematized.

We here summarize the available data on the role of DAXX in mammalian oogenesis and embryogenesis in the context of modern concepts of the chaperone functions of this protein and the molecular mechanisms underlying these functions. In this review, we focus on the importance of DAXX for chromatin reprogramming during development. The cytoplasmic functions of DAXX are not discussed here.

## 2. DAXX Takes the Stage: A Brief History of Its Discovery

The history of DAXX begins in 1997, when Yang and his co-workers were able to identify a novel murine signaling protein, the C-terminal part of which specifically binds to the Fas death domain and enhances Fas-mediated apoptosis [12] by activating the Jun NH_2_-terminal kinase (JNK) pathway through the apoptosis signal-regulating kinase 1 (ASK1) [13]. In parallel, it has been shown that human DAXX (hDAXX), although specifically affecting Fas-mediated apoptosis, does not bind Fas and instead is found in the nucleus [14] due to the presence of two nucleus localization signals (NLS) in the molecule [15].

The structure of the DAXX molecule is now well established. It exhibits six regions of sequence conservation, including a central histone-binding domain (HBD) [4], an N-terminal 4-helix bundle, and a C-terminal domain that is mostly disordered [16]. A combination of structural, biochemical, and cell-based targeting analyses of the H3.3/H4/DAXX HBD complex allowed revealing an extended fold of the DAXX HBD that envelops an H3.3/H4 dimer with seven consecutive α-helices [17,18]. Moreover, the N- and C- termini of the DAXX molecule contain two SUMO-interacting motifs (SIMs: SIM-N and SIM-C, respectively), which both independently interact with a small ubiquitin-related modifier SUMO [19]. A modular structure of the DAXX molecule and its main domains are depicted in Figure 1a.

In the nucleus, SUMO-1, which modifies the promyelocytic leukemia protein (PML), is necessary for the recruitment of DAXX into the PML nuclear bodies (PML-NBs) [22]. The PML-NBs, also known as the nuclear domains 10 (ND10), PML oncogenic domains (PODs) or Kremer bodies, play a role in regulation of transcription and support stability of the genome by sequestration, modification, and/or degradation of nuclear proteins [25]. Nuclear localization of DAXX and colocalization/interaction of DAXX with the PML protein in PML-NBs was confirmed in a yeast two-hybrid screen [26]. In the work mentioned, the repressive function of DAXX to basal transcription, likely by recruiting histone deacetylases, was documented. The repressive activity of DAXX was shown to be regulated by interactions with PML and is inhibited by overexpression of PML.

Already in an early study [23], DAXX was shown to interact with the centromere protein C (CENP-C), and this interaction is mediated by the N-terminal 315 amino acids of CENP-C and the C-terminal 104 amino acids of hDAXX. At the same time, hDAXX was not found predominant in fractions obtained from partially purified mitotic chromosomes, indicating that hDAXX is either not a chromosomal protein during mitosis or that it readily dissociates from condensed mitotic chromosomes during biochemical treatments.

The next important step in DAXX studies was the discovery of functional interactions between DAXX and the chromatin-remodeling protein ATRX—an ATP-dependent helicase, mutations in the gene of which cause the X-linked mental retardation syndrome associated with α-thalassemia [6]. Studies on HeLa cells using co-immunoprecipitation and gel filtration assays have shown that ATRX and DAXX are both the components of the same ATP-dependent chromatin remodeling complex and localize to the PML-NBs. In addition, the level of these DAXX/ATRX complexes is reduced in cells of patients with ATRX syndrome [27]. Later on, Svadlenka and co-authors [21] showed that DAXX can interact with the transcription activator BRG1, another chromatin-remodeling protein with an ATPase activity. They have also documented that DAXX preferentially binds the region between the N-terminal domains QLQ and HSA of BRG1 and weakly interacts with the C-terminal part, suggesting that DAXX can serve as a negative regulator of several BRG1-regulated genes: either directly through BRG1 binding and/or indirectly via other factors. It is possible that BRG1 and ATRX play a role in targeting of DAXX to specific chromatin regions, where DAXX performs its chromatin- and transcription-regulating functions.

## 3. DAXX Is an H3.3 Chaperone

Mammalian H3.3 is a variant of the major histone H3 (H3.1) that differs by only five amino acids [17]. The first studies of H3.3 showed that it is deposited at sites of active transcription with the participation of the histone cell cycle regulator A (HIRA) [28,29]. HIRA is a H3.3 chaperone, the specific function of which has yet to be determined. Later studies showed that H3.3 is deposited in the heterochromatic regions, including telomeres and pericentromeric zones, with participation of the DAXX/ATRX complex [4,30,31,32,33]. These studies provide evidence that DAXX is also an H3.3-specific chaperone facilitating H3.3 deposition at H3K9me3-containing heterochromatin regions. DAXX performs this function in cooperation with ATRX and the highly conserved N-terminus of DAXX interacts directly with variant-specific residues in the H3.3 core [4]. In the absence of DAXX, the replication-dependent Chromatin Assembly Factor 1 (CAF-1) can recruit H3.3 into chromatin, but the deposition pattern of H3.3 changes [31], suggesting that the balance of activities in various histone chaperones affects deposition of the histone variant.

Moreover, DAXX interacts directly with the H3.3/H4 heterodimer through its highly conserved HBD, which plays a critical role in this interaction. The residues E225 in DAXX and G90 in H3.3 are the main determinants of chaperone-mediated H3.3 recognition specificity [17,18]. Two biochemically distinct DAXX-containing complexes can be distinguished: the DAXX/ATRX complex, which is involved in gene repression and maintains the telomeric chromatin structure, and the DAXX-SETDB1-KAP1-HDAC1 complex, which represses endogenous retroviral (ERV) sequences regardless of the inclusion of ATRX and H3.3 into chromatin [34]. Recombinant DAXX assembles H3.3/H4 tetramers on DNA templates, and the DAXX/ATRX complex catalyzes the deposition and remodeling of H3.3-containing nucleosomes [4]. DAXX also uses a shallow hydrophobic pocket to accommodate the small hydrophobic A87 of H3.3, whereas a polar binding environment in DAXX prefers G90 in H3.3 over the hydrophobic M90 in H3.1. The H3.3–H4 heterodimer is bound by the histone-binding domain of DAXX, which makes extensive contacts with both H3.3 and H4 [18].

Importantly, DAXX can specifically associate with H3.3/H4 despite a high concentration of nearly identical canonical H3 in the cell. The mechanism of specific interaction between DAXX and H3.3 was deciphered by DeNizio et al. [24] with the use of a hydrogen/deuterium exchange method coupled to mass spectrometry (H/DX-MS). While most histone chaperones share a globular β-sheet core, which is crucial for histone binding [35,36], the HBD of DAXX is largely disordered in the solution. Formation of the H3.3/H4/DAXX complex induces protein folding and global stabilization of both the histone and its chaperone (Figure 1b). According to the domain-level cooperative folding model [24], a mostly unfolded DAXX initially makes contacts near the H3.3 specificity region and then can sample a large part of the H3.3 surface before folding into place. The dynamic stability of partially folded intermediates may be responsible for the discrimination of H3.3 from other H3 variants [24]. Thus, DAXX, as a molecular chaperone, uses a new strategy that in an unusual way couples its own folding with the recognition and binding of the substrate. This finding determines the fidelity of DAXX in associating with the H3.3 variant, despite an extensive and nearly identical binding surface on its counterparts—H3.1 and H3.2.

DAXX- and ATRX-mediated H3.3 chromatin assembly is required for many functions including H3K9 tri-methylation at pericentromeres, ERV sequences, imprinted genes, intragenic methylated CpG islands and telomeres in embryonic stem cells (ESCs) [37,38,39,40]. The function of DAXX as a H3.3 chaperone is mediated by its interaction with the constitutive centromeric protein CENP-B, which serves as a kind of ‘beacon’ for H3.3 incorporation [41]. Expression of the SUMO-specific proteases results in the removal of DAXX from centromeric regions. The interaction of DAXX with CENP-B and the association of DAXX with centromeres are SUMO-dependent and require two SIMs of the DAXX molecule, enabling DAXX binding to SUMOylated proteins, such as PML [41]. Depletion of SUMO-2, but not SUMO-1, decreases the interaction between DAXX and CENP-B and impairs the accumulation of DAXX and H3.3 at centromeres, which proves the different functions of the SUMO paralogs in the H3.3 deposition processes [41]. Recent data suggest that DAXX-mediated H3.3 deposition is repressed by the PML protein, and PML-NBs coordinate this process [42]. Specifically, PML plays a role in the routing of H3.3 to chromatin and in the organization of megabase-size heterochromatic PML-associated domains (PADs). Loss of PML impairs the heterochromatic state of PADs by shifting the balance of H3 methylation from K9me3 to K27me3. In addition, this alters the ATRX/DAXX-dependent, but not HIRA-dependent, deposition of H3.3 in PADs [43].

It can be assumed that DAXX not only regulates the deposition of H3.3, but is one of the leading factors in maintaining the global heterochromatin landscape, since loss of DAXX can seriously affect the subnuclear organization in general. In particular, the use of electron spectroscopic imagine (ESI)—a specialized form of energy-filtered transmission electron microscopy that allows one to visualize chromatin domains in situ with high contrast and spatial resolution—revealed dramatic changes in the organization of H3K9me3-enriched heterochromatin domains, which occur in the absence of DAXX [44]. Among these changes, the loss of a typical chromocenter structure and the loss of overlap between non-nucleolar and perinucleolar compact chromatin and H3K9me3 were observed. Moreover, a compact chromatin not marked by H3K9me3 was found to be ectopically accumulated, and H3K9me3-enriched domains become uncoupled from major satellite DNA. Furthermore, the increase in H3.3 associated with H3K9me3-enriched chromatin domains and the loss of structural boundaries between heterochromatin and the nucleolus were also documented. In addition, the structural integrity of nucleoli and the organization of rDNA were disrupted [44].

## 4. DAXX is Essential for Normal Development of Oocytes and Embryos

In the shoreless sea of nuclear compounds DAXX emerges as an iceberg, demonstrating its significant role in global transformations of the nucleus during late oogenesis and early embryogenesis, but mostly, its function, which could impact on development, is still elusive. The first attempts to shed light on the role of DAXX in embryonic development began immediately after the identification of this protein. Michaelson et al. [45] were the first who obtained DAXX-deficient mice, expecting to find a hyperproliferative disorder in their development. However, the result was unexpectedly opposite: mutations in the *DAXX* gene led to extensive apoptosis, as indicated by a lot of piknotic nuclei in preparations and TUNEL assay results. The highest levels of apoptosis were found in the allantois and the neuroepithelium [45]. It has been found that post-implantation DAXX^−/−^ embryos are developmentally retarded, which is the most prominent at 10.5 day of gestation. In the mutant embryos, normal development of the ectoderm and somites is disrupted. In addition, there was no evidence for formation of the placenta, although some extraembryonic structures such as the yolk sac and amnion were observed in histological sections [45]. At 11.5 days of gestation, DAXX^−/−^ embryos started to dissolve and completely disintegrated by the 12.5 day [45,46]. As shown later, total loss of DAXX impairs development even at the pre-implantation stages, since only 50% of DAXX-deficient embryos can reach the blastocyst stage [47].

Since DAXX is an H3.3 chaperone, the disruption of the developmental program in DAXX-deficient embryos is not surprising due to the significant role of H3.3 in oogenesis and embryogenesis. H3.3 knockdown in mouse oocytes leads to impaired reprogramming and suppresses transcription of key genes supporting pluripotency [48]. Soon after fertilization, H3.3 is removed from the maternal pronucleus (mPN) of the zygote, suggesting that the epigenetic marks carried by H3.3 in oogenesis are erased in zygotes [49]. It has also been shown that in mouse zygotes, H3.3 plays an important role in proper formation of the paternal pronucleus (pPN) and significantly affects further development of the embryo (see [10] and references therein). It should be noted that H3.3 deposition has a replication-independent mechanism, as shown by microinjection experiments with mRNAs encoding FLAG-tagged H3.3 [50], whereas the canonical histone H3 is included only during DNA replication [51].

The whole-transcriptome analysis of metaphase II (MII) oocytes allowed revealing expression of DAXX and other H3.3 chaperones, such as HIRA, in mouse oogenesis [52]. However, DAXX cannot compensate normal H3.3 deposition in HIRA-depleted oocytes [53]. Similar results were also obtained for mouse ESCs [32].

In oocytes, DAXX predominantly localizes to the pericentromeric heterochromatin (PCH) regions, as shown by analysis of the nuclear localization of DAXX and centromeric proteins [54]. It has recently been shown that DAXX deposition in PCH is significantly reduced in fully-grown GV (germinal vesicle stage) oocytes deficient in the heterochromatin protein 1β (HP1β), also known as the Chromobox Protein Homolog 1 (CBX1), and completely abrogates in oocytes deficient of the methyltransferase SUV39h2, in which the PCH regions lack both H3K9me3 and HP1β [47]. After fertilization, DAXX is revealed in the pronuclei of zygotes, initially in close association with the so-called nucleolus precursor bodies (NPBs) [47,55,56]. The NPBs are peculiar atypical nucleoli that serve as major organizing structures for heterochromatin in mammalian zygotes and early embryos [57], including pericentromeric chromatin [58]. Interestingly, DAXX distribution in paternal PCH exhibits a dotted pattern, described as a “pearls on a string” structure, while the DAXX signal is more continuous in maternal PCH [47]. The general dynamics of DAXX in the nucleus during mammalian development, from the GV stage oocyte to morula, is shown schematically in Figure 2.

In GV-stage oocytes, DAXX is concentrated predominantly in several large chromocenters. In zygotes, DAXX distribution patterns are different in the paternal (pPN) and maternal pronucleus (mPN). In the pPN, DAXX is concentrated in small dotted areas that presumably correspond to pericentromeric heterochromatin at the periphery of the atypical nucleoli, or nucleolus precursor bodies (NPBs). In the mPN, DAXX distribution is more continuous in NPB-associated heterochromatin regions. At the late two-cell stage, many NPBs exhibit prominent DAXX-containing areas at their periphery. Some large DAXX-positive typical chromocenters, not associated with the NPBs, are also detected. Areas of high local concentration of DAXX almost disappear in morula nuclei, with the exception of a few that are still associated with the nucleolar periphery in some blastomeres. In Figure 2, the oocyte nucleus (GV), zygotic pronuclei, and blastomere nuclei are presented by circles; the cytoplasm is not shown.

The experiments, when the atypical nucleoli were removed from immature oocytes (enucleolation), have shown that the components of the maternal “nucleolus” are critically important for the deposition of DAXX in zygotes [55]. The absence of maternal “nucleoli” affects the dynamics of the S phase, and DAXX is no longer detected in enucleolated zygotes, suggesting that DAXX is not stably associated with maternal DNA.

As in somatic cells, H3.3 deposition in mammalian oocytes and embryos involves DAXX in the complex with ATRX. Moreover, both these proteins could operate in different complexes and assemble on chromatin with different kinetics [47,59]. The ATRX protein is necessary to recruit DAXX to PCH during meiotic prophase I. In the absence of ATRX, the DAXX protein fails to associate with the PCH regions in 81.3% of oocytes at the GV stage, despite of these regions containing H3K9me3, a mark of repressed chromatin [54]. In contrast with somatic cells in which the association of DAXX with ATRX at the PCH regions exists only for a brief period at the S phase [27,46], the DAXX/ATRX complex remains in association with these heterochromatin regions while an oocyte grows [54].

It should be noted that oocytes are highly specialized cells that demonstrate some unusual features compared to somatic cells in many ways. Most importantly, a growing oocyte is in meiotic prophase, at the prolonged diplotene stage. Mammalian diplotene (dictyate) oocytes are arrested at a stage equivalent to the G2/M transition in somatic cells. A physiologically determined suppression of chromatin activity occurs at this stage, which is often manifested in the formation of a heterochromatin special structure in the GV, termed the karyosphere or karyosome [60]. The karyosphere formation is essential in the establishment of unique chromatin remodeling events in the oocyte genome [10,54]. Thus, the S phase-independent interaction between DAXX and ATRX in oocytes may be important to maintain the highly condensed state of the karyosphere, which is essential for the oocyte to develop properly.

After fertilization, а clear colocalization of DAXX and ATRX initially revealed already at stage PN0 in the mPN and at stage PN2 in the pPN, i.e., before replication, but thereafter, the DAXX level reduces rapidly in maternal PHC [47]. The most pronounced colocalization areas of DAXX and ATRX are observed in the nucleus of mouse late two-cell embryos. However, the heterochromatin regions are also observed, in which only one protein from the pair is localized [56], which, at first glance, contradicts the well-established concept of the critical role of the ATRX–DAXX interaction in heterochromatin remodeling. In our opinion, this contradiction could be explained by the specific features of early embryos—in particular, by the processes of the zygotic genome activation (ZGA) [61]. In the mouse, the main ZGA events complete by the end of the two-cell stage. During this period, maternal regulatory molecules largely degrade and are replaced by molecules of zygotic origin [62]. In this context, the chromatin regions in which DAXX and ATRX are separately detected may be the storage sites for the degrading proteins of maternal origin.

Importantly, the character of functional interactions between DAXX and ATRX after fertilization is fundamentally different in pPNs and mPNs, indicating a known asymmetry of the pronuclei in mammalian zygotes [47]. In the mPN, DAXX colocalizes with ATRX in the PCH regions of decondensed chromosomes containing H3K9me3 and HP1β. However, in the pPN, these chromatin regions contain only DAXX but not ATRX. It was also demonstrable that DAXX is necessary for ATRX recruitment to the PCH regions in paternal but not maternal pronuclei. In contrast, ATRX is required in the mPN to recruit DAXX, as shown in experiments with ATRX-depleted oocytes [47].

Complementation assays showed that DAXX-mediated H3.3 deposition is required for chromosome stability in mouse early embryos. The DAXX protein was shown to regulate repression of the Polycomb Repressive Complex 1 (PRC1) target genes in oogenesis and early embryogenesis. However, PRC2 and H3K27me3 do not serve as key determinants of DAXX recruitment and function in mouse zygotes [47]. The role of DAXX in maintaining the genome integrity is also evident during the period of global DNA demethylation that occurs after fertilization [63] and is accompanied by an increase in the recruitment of the DAXX/ATRX complex to tandemly repeating sequences, including retrotransposons and telomeres [38]. Knockdown of DAXX/ATRX in cells with hypomethylated DNA increases the aberrant derepression of tandem repeat transcription and expands telomere dysfunction. The DAXX/ATRX complex also suppresses H3K9 tri-methylation during mouse embryogenesis, interacting with the methyltransferase SUV39h [38].

As an H3.3 chaperone, DAXX is probably involved in remodeling of pericentromeric heterochromatin—the main place where H3.3 is deposited in oocytes and embryos. It has been shown that the mutation K27R in the H3.3 molecule results in the aberrant accumulation of pericentromeric transcripts, HP1 mislocalization, dysfunctional chromosome segregation, and developmental arrest [64]. Together with ATRX, the DAXX protein is also involved in repression of telomeric sequences. A central role for DAXX in association of the DAXX/ATRX complex with telomere/subtelomere regions was confirmed in experiments using mouse ESCs knocked out for DNA methyltransferases [38]. Loss of ATRX has a minor effect on the telomeric localization of DAXX, but knockdown of DAXX severely compromised the ability of ATRX to localize to telomeres. As DAXX and ATRX are localized in the telomeric regions of embryonic chromosomes, it cannot be excluded that in embryogenesis, DAXX and ATRX are involved in the Alternative Lengthening of Telomeres (ALT), especially since mutations in the *DAXX* gene have been described in patients with telomerase-negative pancreatic neuroendocrine cancers [65].

To discuss experimentally proven and potential functions of DAXX in developmental processes, it is necessary to note the importance of this protein for the regulation of the cell cycle, mitosis, and cytokinesis. A decrease in the S-phase duration accompanied by an increase in the number of cells with double nuclei, micronuclei, and nuclear blebs was observed in DAXX^−/−^ cells [41,46], indicating mitotic segregation defects. Although these observations were made on cultured cells, the authors believe that it is the disruption of the normal coordination of the cell cycle after DNA replication and incomplete mitosis, which might be the basis for the developmental problems observed in DAXX knockout animals [46]. DAXX can be involved in the control of mitosis in particular through the regulation of Aurora-A kinase and cyclin B stability, including through the interaction of DAXX with the mitotic checkpoint protein RASSF1 [66,67]. It should be verified whether these mechanisms established for cultured DAXX knockout cells can be implemented in oocytes and preimplantation embryos.

Somatic cell studies also provide evidence that DAXX plays an important role in the regulation of centromeric heterochromatin activity. It has been found that knockout of DAXX reduces the association of ATRX with centromeres [46] and significantly increases chromosomal instability, which is consistent with mitotic abnormalities and micronuclei formation observed in these cells [41]. Accumulation of DAXX in centromeric and pericentromeric chromatin regions is enhanced by stress, including heat shock, as shown for several cancer cell lines [68], suggesting that DAXX-containing complexes are useful in maintaining the normal epigenetic landscape of heterochromatin. Given the high dynamics of the molecular composition and spatial organization of centromeric/pericentromeric heterochromatin in oocytes and embryos [10], one can assume that DAXX performs similar functions in mammalian development.

Thus, it should be noted that even the scanty data available in the literature on the role of DAXX in oogenesis and early embryogenesis indicate that this protein is an important factor in proper development. Loss of DAXX in oocytes and embryos affects many nuclear processes (Figure 3). In oocytes, zygotes, and early embryos, DAXX functions in close cooperation with other proteins involved in chromatin remodeling, as in somatic cells. These functional interactions are critical for normal DAXX deposition in the heterochromatin compartment during mammalian development (Table 1).

## 5. Conclusions

There are relatively scarce data on the dynamics and molecular interactions of DAXX in mammalian oogenesis and embryogenesis compared to those obtained in in vitro studies on cultured cells. This is explainable, since most of the methods of molecular genetic and biochemical analysis are difficult to adapt to oocytes and/or embryos. At the same time, the available data allow us to conclude that the functional interactions of DAXX in developmental processes may have specific features and differ from those in tissue-culture somatic cells. This specificity is easily understandable, given the unique character of the cell cycle at the final stages of oogenesis and at the preimplantation stages of development. In our opinion, data on the asymmetry of functional relationships in the DAXX/ATRX pair in the male and female pronuclei of zygotes are of particular interest. Traditionally, ATRX is considered as a leader in this complex, recruiting DAXX to targeting chromatin zones, since the ATRX protein can interact not only with H3K9me3 and HP1, but also directly with structured DNA, such as G-quadruplexes [69]. However, taking into account the results presented in the paper by Liu et al. [47], according to which, on the contrary, DAXX recruits ATRX in the pPN, it can be assumed that DAXX functions in embryogenesis are broader than what is assumed for somatic models. The question also remains how the distribution of DAXX between heterochromatin and the interchromatin space of the nucleus is regulated. The PML-NBs play a leading role in this process in the nucleus of somatic cells [22], but we are not aware of any data on the presence of these nuclear domains in oocytes or pre-implantation embryos of mammals. We hope that that further improvement of single cell technologies will provide a more complete understanding of the role of DAXX in development of mammals, including human.

## Figures and Tables

**Figure 1 ijms-22-01313-f001:**
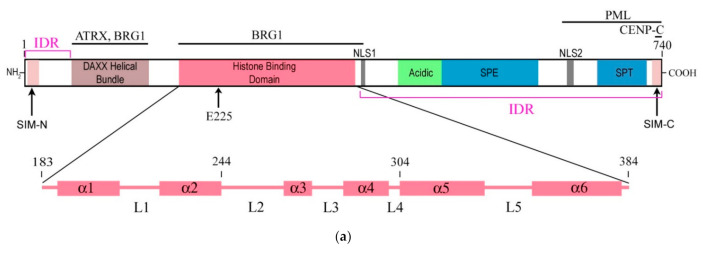
Simplified diagrams depicting the modular structure of the DAXX molecule (**a**) and interactions between the histone binding domain of DAXX and H3.3 during the formation of the complex H3.3/H4/DAXX (**b**). (**a**) DAXX contains two independent SUMO-interacting motifs at the N- and C-termini (SIM-N and SIM-C, respectively) [19]. The DAXX helix bundle serves as the main site for binding the α-thalassemia/mental retardation syndrome X-linked protein ATRX [20] and many other proteins that interact with the DAXX terminal region, including BRG1; another BRG1-interacting region is in the central part of DAXX [21]. The histone binding domain (HBD) is the main module involved in interactions with H3.3, a functional partner of DAXX, and the residue E225 in DAXX contributes to the specificity of this interaction. The HBD consists of six α-helices (α1–α6) and intervening loops (L1–L5); N-terminal helices (α1 and α2) are also termed ‘tower’ [17]. Two nucleus localization signals (NLS1 and NLS2), a Glu/Asp-rich acidic region, two segments rich, respectively, in Ser/Pro/Glu residues (SPE) and in Ser/Pro/Thr residues (SPT) are depicted. The most intrinsically disordered regions (IDRs) of the DAXX molecule are also shown. The C-terminus includes the regions to interact with the PML protein [22] and CENP-C [23], which impacts the peculiarities of DAXX distribution within the nucleus. Please note that the vast majority of DAXX interactions, not mentioned in our review, including those involved in cancer, are not depicted in the scheme (e.g., see [8] for more comprehensive list of references). (**b**) DAXX is a specific H3.3 chaperone. The recognition and binding of H3.3 by the HBD of DAXX involve several folding steps [24], shown in the diagram by encircled numbers (1–5). 1—L1 of the DAXX ‘tower’ interacts with lateral H3.3 surface by long-range electrostatic interactions; 2–the ‘tower’ helices (α1 and α2) of DAXX, including portions of L1, fold onto two helical segments of H3.3 (α1 and α2), including the intervening loop (the interacting domains are ensquared); 3—the αN helix of H3.3 folds; 4—the L2 of DAXX tightly wraps around the folded H3.3 αN helix; 5—finally, the α5 and α6 helices of DAXX pack against H3.3 and H4, respectively, forming the H3.3/H4/DAXX complex. H4 interactions are not shown. For further reading and structural details regarding formation of the H3.3/H4/DAXX heterotrimer, see [17,24].

**Figure 2 ijms-22-01313-f002:**
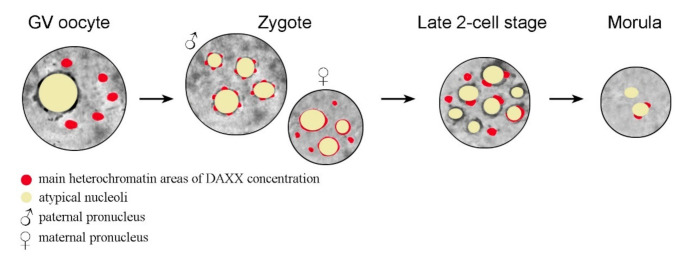
A cartoon illustrating the dynamics of DAXX distribution in the heterochromatin compartments during mouse early development.

**Figure 3 ijms-22-01313-f003:**
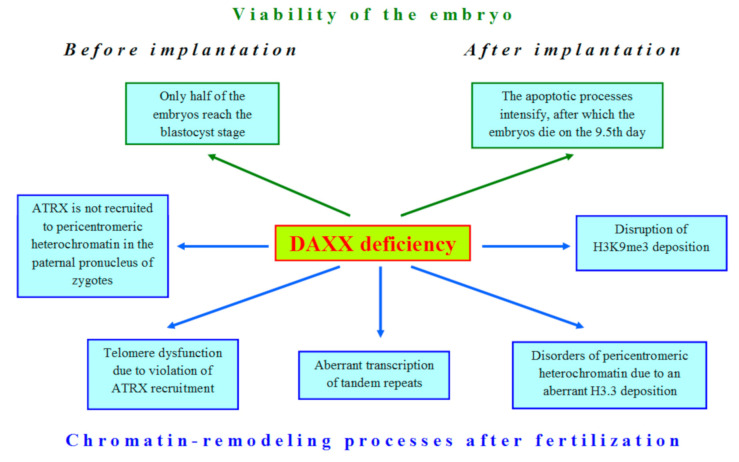
The deficiency of DAXX violates many cellular processes in early development of mammals.

**Table 1 ijms-22-01313-t001:** Representative factors that regulate DAXX deposition in mammalian oocytes and zygotes.

Factor	Consequent Effect of Deficiency	Stage	References
HP1β (CBX1)	Significant reduction of DAXX deposition in pericentromeric heterochromatin	Fully-grown oocytes	[47]
SUV39h2	Complete abrogation of DAXX deposition in pericentromeric heterochromatin	Fully-grown oocytes	[47]
Maternal nucleolar proteins	Absence of DAXX in enucleolated zygotes	Zygotes	[55]
ATRX	Violation of DAXX recruitment to pericentromeric heterochronatin	Meiotic prophase I	[54]
	Loss of DAXX in the maternal pronucleus	Zygotes	[47]

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
