# Peer review of "DAXX Is a Crucial Factor for Proper Development of Mammalian Oocytes and Early Embryos"

_ijms, 2021, doi:10.3390/ijms22031313_

Round 1

Reviewer 1 Report

The manuscript by Bogolyubov and Bogolyubov reviews the current knowledge on the role of the protein DAXX in chromatin remodelling, in particular, in association with the helicase ATRX. The manuscript provides a lot of information and refers to many publications available on the topic. However, there is a general lack of critical integration of the available informations, significantly reducing the interest of the manuscript. Moreover, although the title suggests that the review will cover the role of DAXX in oocytes and embryos, the main focus of the manuscript is actually on the function of DAXX in chromatin remodelling, and only marginally on its developmental functions. Although as the authors mention the data available on the topic is limited, if reviewing the function of DAXX during development all aspects should have been covered among others its implication in mitosis and cytokinesis, a description of the developmental defects of DAXX mutants, DAXX cytoplasmic localisation. Moreover known differences between oocytes and somatic cells could be discussed more extensively. For exemple the interaction between DAXX and ATRX beyond S-phase in oocytes  which is mentioned in lines 260-61. How would this help chromatin remodelling during oogenesis? 

Figures could also be improved to help understanding.

Figure 1 is currently very small. The authors could include a second panel with a model depicting the DAXX-H3.3 interactions. This would help explaining their mechanism of interaction and clarify its specificity, as the authors mention in section 3 (lines146-152) that DAXX uses a non canonical strategy for binding to H3.3 and for its chaperone function. It would also be useful to describe the canonical mechanism and to highlight the differences, especially for non specialist readers.

Figure 2 and 4 are not very informative and do not help understanding of the text. One of them could be substituted with a model of DAXX interactions and mode of function.

Figure 3 is not necessary as it shows that DAXX and ATRX do not always co-localize in mouse 2-cell embryos (already published data from the authors) but the relevance of this lack of co-localization is not discussed. 

Minor points:

  1. In figure 1 the labels are very small and difficult to read. The authors should increase the font size. They should also use the same abbreviations/terminology as in text. In the text, SIM1 and SIM2 are called SIM-N and SIM-C. Interaction site with BRG1 could also be included as it is described in the text. Similarly important residues which are mentioned in the text, such as G90, could be indicated in the model.
  2. If keeping figure 2, the authors should make it clear that the circles are nuclei and not cells. 
  3. In the legend to Figure 2 the authors report a percentage. It is not clear what it indicates: variation in surface, number of dots, amount of protein? This should be clarified.
  4. lines 298-299 are not in English.

Author Response

We thank Reviewer 1 for his/her critique and valuable comments.

Referee's suggestion: The manuscript by Bogolyubov and Bogolyubov reviews the current knowledge on the role of the protein DAXX in chromatin remodelling, in particular, in association with the helicase ATRX. The manuscript provides a lot of information and refers to many publications available on the topic. However, there is a general lack of critical integration of the available informations, significantly reducing the interest of the manuscript. Moreover, although the title suggests that the review will cover the role of DAXX in oocytes and embryos, the main focus of the manuscript is actually on the function of DAXX in chromatin remodelling, and only marginally on its developmental functions. Although as the authors mention the data available on the topic is limited, if reviewing the function of DAXX during development all aspects should have been covered among others its implication in mitosis and cytokinesis, a description of the developmental defects of DAXX mutants, DAXX cytoplasmic localisation. Moreover known differences between oocytes and somatic cells could be discussed more extensively. For exemple the interaction between DAXX and ATRX beyond S-phase in oocytes which is mentioned in lines 260-61. How would this help chromatin remodelling during oogenesis?

Response. We add to the text a brief discussion of the importance of Daxx for mitosis and cytokinesis based on the results of experiments with Daxx depletion in cultured somatic cells (lines 359–371). Unfortunately, we are not aware of similar data obtained on cleavage embryos. We also briefly discuss the importance of DAXX in maintaining the epigenetic status of centromeric heterochromatin (lines 372–382) and provide information on the developmental defects of DAXX mutants in the revised text (lines 216–225). The cytoplasmic localization of DAXX is not discussed, because we would like to focus especially on the importance of DAXX for chromatin reprogramming during development. Finally, we agree that oocytes are specialized cells strictly different from somatic cells. In particular, mammalian oocytes are arrested in meiotic prophase (at diplotene stage) — a stage equivalent to the G2/M transition in somatic cells. The S phase-independent interaction between DAXX and ATRX at this stage may be important to maintain the highly condensed state of chromatin. Changes are made in the text (lines 288–300).

Figures could also be improved to help understanding.

Figure 1 is currently very small. The authors could include a second panel with a model depicting the DAXX-H3.3 interactions. This would help explaining their mechanism of interaction and clarify its specificity, as the authors mention in section 3 (lines146-152) that DAXX uses a non canonical strategy for binding to H3.3 and for its chaperone function. It would also be useful to describe the canonical mechanism and to highlight the differences, especially for non specialist readers.

Response. We expand Figure 1 to show alpha-helical structure of the DAXX HBD (Figure 1a) and interactions between the HBD and H3.3 during formation of the complex H3.3/H4/DAXX (Figure 1b). In the text, we briefly discuss the domain-level cooperative folding model (DeNizio et al., 2014) that describes the mechanisms of specific H3.3 recognition by DAXX in the context of the largely disordered structure of its molecule. This feature distinguishes DAXX from other histone chaperones (lines 158–169).

Figure 2 and 4 are not very informative and do not help understanding of the text. One of them could be substituted with and mode of function.

Response. We would like to keep these figures, since they are directly related to the main topic of the review and, in particular, reflect the specific morphological features of the nuclei of oocytes and early embryos. A model of DAXX interactions with H3.3 is shown in Figure 1b.

Figure 3 is not necessary as it shows that DAXX and ATRX do not always co-localize in mouse 2-cell embryos (already published data from the authors) but the relevance of this lack of co-localization is not discussed.

Response. We briefly discussed this point in the text (lines 306–314).

Minor points:

    In figure 1 the labels are very small and difficult to read. The authors should increase the font size. They should also use the same abbreviations/terminology as in text. In the text, SIM1 and SIM2 are called SIM-N and SIM-C. Interaction site with BRG1 could also be included as it is described in the text. Similarly important residues which are mentioned in the text, such as G90, could be indicated in the model.

Response. Figure 1 is changed according to Referee’s suggestions.

    If keeping figure 2, the authors should make it clear that the circles are nuclei and not cells.

Response. We clarify this in the legend (Lines 271–272).

    In the legend to Figure 2 the authors report a percentage. It is not clear what it indicates: variation in surface, number of dots, amount of protein? This should be clarified.

Response. We mean the percentage of NPBs at the periphery of which DAXX-positive regions can be localized. To avoid confusion, we remove this issue from the legend.

    lines 298-299 are not in English.

Response. Oh, sorry for this pratfall.

Reviewer 2 Report

Dear authors,

amazing review!! I really enjoyed learning about DAXX. 

DAXX is a promising source of work.

Thank you so much for such rich work.

Best regards

Author Response

We are grateful to Reviewer 2 for the high assessment of our work. Nesessary corrections are made in the text.

Round 2

Reviewer 1 Report

The revised version of the manuscript by Bogolyubov and Bogolyubov is a thorough summery of the literature available on DAXX in chromatin remodelling with a slight focus on the role of this protein in oocytes and embryos, due to the paucity of data available on the topic. The authors have addressed most of the individual issues in the original review, however I think the narrative is still convoluted.

Figure 1 has been changed, as asked, and it is now very complex but not entirely clear, particularly the final scheme in Figure 1b, step 5. It should be changed or removed. As a minor point, the labels a- and b- to Figure 1 are in the wrong place.

The authors have also decided to keep Figure 2 and 3. Some text has been added to explain their relevance but I still find them uninformative.

A final minor point. In the new version of the manuscript the authors write that information derived from GV studies may be not generally relevant. Although I understand where the authors are coming from, I find the statement extremely disturbing in that specific functions of proteins in oocyte are very relevant to understanding the analysed process. Lines 289-290 (As a result,.....not generally relevant ) can be removed to avoid misunderstanding on the importance of studying non generalisable mechanisms, without affecting the manuscript.

Author Response

Dear Reviewer,

Thank you for the comments.

  1. Suggestion:

The revised version of the manuscript by Bogolyubov and Bogolyubov is a thorough summery of the literature available on DAXX in chromatin remodelling with a slight focus on the role of this protein in oocytes and embryos, due to the paucity of data available on the topic. The authors have addressed most of the individual issues in the original review, however I think the narrative is still convoluted.

Figure 1 has been changed, as asked, and it is now very complex but not entirely clear, particularly the final scheme in Figure 1b, step 5. It should be changed or removed. As a minor point, the labels a- and b- to Figure 1 are in the wrong place.

Response. We replace the final scheme in Figure 1b with a verbal description; a- and b- parts of the figure are formatted according to the template of the journal.

  1. Suggestion:

The authors have also decided to keep Figure 2 and 3. Some text has been added to explain their relevance but I still find them uninformative.

Response. Figure 3 is removed.

  1. Suggestion:

A final minor point. In the new version of the manuscript the authors write that information derived from GV studies may be not generally relevant. Although I understand where the authors are coming from, I find the statement extremely disturbing in that specific functions of proteins in oocyte are very relevant to understanding the analysed process. Lines 289-290 (As a result,.....not generally relevant ) can be removed to avoid misunderstanding on the importance of studying non generalisable mechanisms, without affecting the manuscript.

Response. The phrase “As a result, it is sometimes concerned that information derived from the GV studies may not be generally relevant” is removed, together with the reference [60].